# City-Wide Eco-Routing Navigation Considering Vehicular Communication Impacts

**DOI:** 10.3390/s19020290

**Published:** 2019-01-12

**Authors:** Ahmed Elbery, Hesham Rakha

**Affiliations:** Center for Sustainable Mobility, Virginia Tech Transportation Institute, Blacksburg, VA 24061, USA; aelbery@vt.edu

**Keywords:** ITS, VANET, eco-routing, large-scale network, smart cities, penetration ratio, connected vehicles, vehicular networks

## Abstract

Intelligent Transportation Systems (ITSs) utilize Vehicular Ad-hoc Networks (VANETs) to collect, disseminate, and share data with the Traffic Management Center (TMC) and different actuators. Consequently, packet drop and delay in VANETs can significantly impact ITS performance. Feedback-based eco-routing (FB-ECO) is a promising ITS technology, which is expected to reduce vehicle fuel/energy consumption and pollutant emissions by routing drivers through the most environmentally friendly routes. To compute these routes, the FB-ECO utilizes VANET communication to update link costs in real-time, based on the experiences of other vehicles in the system. In this paper, we study the impact of vehicular communication on FB-ECO navigation performance in a large-scale real network with realistic calibrated traffic demand data. We conduct this study at different market penetration rates and different congestion levels. We start by conducting a sensitivity analysis of the market penetration rate on the FB-ECO system performance, and its network-wide impacts considering ideal communication. Subsequently, we study the impact of the communication network on system performance for different market penetration levels, considering the communication system. The results demonstrate that, for market penetration levels less than 30%, the eco-routing system performs adequately in both the ideal and realistic communication scenarios. It also shows that, for realistic communication, increasing the market penetration rate results in a network-wide degradation of the system performance.

## 1. Introduction

Intelligent Transportation Systems (ITSs) use networked sensors, microchips, and communication technologies to collect, process, and disseminate information about the state of the transportation system. Using this data, the traffic management center (TMC) can improve the performance of the overall transportation system by making better informed decisions which can reduce travel time and fuel consumption, and mitigate traffic congestion. These decisions are affected by the accuracy, completeness, and spatial distribution of the collected data, which is influenced by the communication system. Vehicular Ad-hoc Networks (VANETs), which are standardized in [1], are expected to constitute the ITS communication infrastructure. Consequently, the performance of an ITS is significantly influenced by the VANET communication performance in terms of packet drop rate and packet delay. Thus, it is imperative to study the impact of VANET communication on the performance of transportation applications. This impact is dependent on the ITS application itself and its sensitivity to the completeness, correctness, and the spatial distribution of the data. One of these applications is feedback-based eco-routing (FB-ECO) navigation [2,3,4].

The FB-ECO navigation is an ITS application that aims to minimize the average vehicle’s fuel consumption and emission levels, by routing vehicles through the most environmentally friendly routes. It utilizes connected vehicle technology to collect real-time fuel consumption information from probe vehicles to compute the best routes. FB-ECO navigation system assumes the capability of some vehicles (known as sensor vehicles or probe vehicles) to compute the fuel consumption on each traversed road link, which implicitly means these probe vehicles are equipped with Global Positioning Systems (GPSs). It also assumes that these probe vehicles are connected to TMC, to which they report the computed fuel consumption and the associated road links. Subsequently, all vehicles can get dynamic route guidance, when needed.

In a vehicular environment, there is a bidirectional impact between communication and mobility. It is well-known that communication network performance can be affected by mobility in the transportation network (i.e., congestion levels and speeds). On the other hand, in the FB-ECO navigation systems, communication performance (i.e., packet drop rate and packet delay) can also affect vehicle routing decisions and, consequently, vehicle mobility. For example, a higher packet drop rate leads to a lack of link cost updates at the TMC, causing the TMC to route vehicles through sub-optimal routes, resulting in higher fuel consumption. Another example, which is an important bidirectional interaction between communication and transportation, is that the packet drop rate becomes higher and the delay becomes longer in congested road networks, which leads to losing more updating packets and a resultant larger deviation from the best routes.

This bidirectional interdependency creates a loop of mutual influence between communication and transportation systems [5] that increases the complexity (and, consequently, the analysis) of these systems. In fact, studying and modeling such systems is challenging, not only because of this interdependency of the communication and the mobility components, but also because of the scale at which these systems are applied—covering a city-level road network.

In addition to the impact of communication, the market penetration ratio (MPR) of the eco-routing probe vehicles can also significantly affect the system-wide eco-routing performance. A lower market penetration ratio can lead to a lack of real-time link cost information, similar to that of the communication impact. However, there are important differences between the information deficiency in the two cases. The lack of information due to low MPR of the probe vehicles is supposed to be spatially uniformly distributed (based on the assumption that probe vehicles are spatially uniformly distributed), while in the case of communication, the TMC suffers information deficiency in congested areas (and the surrounding areas, because of the long communication range reaching up to 1000 m), where the packets experience higher drop rates and longer delay. Another difference between the two sources of impacts is that the communication impact is not limited to dropping packets only, it also increases the packet delay.

Understanding these impacts of communication and penetration ratio on eco-routing performance is a key enabler to deploying these system. Thus, in this paper, we build upon our previous work in [4,6,7,8] to study the impacts of penetration ratios of the probe vehicles on the performance of the FB-ECO in a large-scale city level road network with real calibrated traffic. Then, we study the impact of communication on the FB-ECO performance in the same network. We conduct this study and compare the eco-routing performance in two main cases:The ideal communication case, that assumes a perfect communication performance (i.e., no packet drops or delay); andthe realistic communication modeling case, in which we model VANET communication and study its impact on the system performance.

We conduct these studies at different vehicle traffic congestion levels.

The remaining of this paper is organized as follows. Section 2 gives an overview of the eco-routing navigation system and its literature. Section 3 describe the implementation and operation of FB-ECO, assuming ideal communication. In Section 4, the communication model is presented and the operation of the FB-ECO with the communication modeling is described. The simulation network and results are presented in Section 5, before the final conclusion.

## 2. Eco-Routing

In the last two decades, the environmental and economical impacts of the transportation sector have attracted the attention of scholarly communities, who devoted much research effort towards sustainable mobility, to save fuel consumption and emissions. Thus, eco-routing navigation techniques were introduced to achieve this objective by utilizing the route fuel cost as a metric, based on which the most environmentally friendly routes are be selected.

Developing and deploying eco-routing navigation techniques is challenging. One major challenge is the estimation of the route fuel cost. This challenge comes from the fact that the route fuel cost is a function of many parameters, including route characteristics (i.e., length, maximum speed, grade), vehicle characteristics (e.g., weight, shape, engine, and power), and driving behavior. So, it is too difficult to combine all these parameters in a single model, especially because many of these parameters are stochastic and there is a complex dependency among all of them. Therefore, the best way to calculate the route cost is to use real-time feedback from vehicles moving on these routes—we call this technique FB-ECO. These data can be collected in real-time and fused with historical data to estimate the route fuel cost and, consequently, calculate the best route for vehicles dynamically and in real-time. More details about eco-routing and FB-ECO can be found in [2,3].

This feedback-based eco-routing system is simple and can accurately and dynamically compute the route cost. But, on the other hand, deploying such a system requires a vehicle’s capability to quantify the fuel consumption for each road link it traverses, which implicitly means that the probe vehicle should be equipped with GPSs. It also requires the communication network, to enable vehicles to communicate the computed costs to the TMC.

### 2.1. Literature of Eco-Routing

Eco-routing was initially introduced in [9], and was applied to the street network in the city of Lund, Sweden, to select the route with the lowest total fuel consumption, and thus the lowest total CO${}_{2}$ emissions. The streets were divided into 22 classes, based on the fuel consumption factor for peak and non-peak hours, and three vehicle classes were used. This routing technique resulted in a 4% average savings in fuel consumption.

Authors in [10] demonstrated the importance of route selection on fuel and the environment. They showed that emission and energy optimized traffic assignments, based on speed profiles, can reduce CO${}_{2}$ emissions by 14% to 18%, and fuel consumption by 17% to 25%. In [11] is another attempt to minimize vehicle fuel consumption and emission levels, by proposing a new set of cost functions which includes fuel consumption and emission levels for the road links. In [3], the authors developed an eco-routing navigation system that uses both historical and real-time traffic information to calculate the link fuel consumption levels, and then selects the fuel-optimum route. In [8] is an attempt to enhance the eco-routing algorithm developed in [10], by introducing a new ant-colony based updating technique for eco-routing. In 2017, we have developed the first system optimum eco-routing model [12], that uses linear programming and stochastic route assignment to minimize system wide fuel consumption. The system optimum eco-routing reduced the fuel consumption by about 36%, compared to the user equilibrium model. In [13], the authors used the model developed in [14] to reduce the transportation-related energy in the city of Los Angeles, influencing individual behavior and multi-modal route selection based on driver preferences by utilizing artificial intelligence and machine learning techniques to compute personalized multi-modal routes.

All of these previous efforts did not consider the communication network and its influence on FB-ECO system performance. The only work that considered this impact is our previous work in [4], which used discrete event simulation to model communication in VANET. However, this system showed limited scalability. So, in [6], we developed a novel communication model and incorporated it within a microscopic traffic simulator, the INTEGRATION software [15], to enable modeling of communication and transportation in large-scale vehicular systems. In this paper, we utilize this communication model to conduct our sensitivity analysis of both communication and penetration ratio and their impacts on eco-routing.

### 2.2. Eco-Routing as a Feedback User-Equilibrium Model

Eco-routing was developed as a user-equilibrium model by using the shortest path techniques. So, it basically tries to minimize individual vehicle fuel consumption. The user-equilibrium model for eco-routing can be defined as follows: Given a road network directed connected graph G(N,L,C), where N={1,2,…,n} is a set of *n* road network nodes, $L=\{l_{ij}:i,j\in N\}$ is a set of *l* directed road links, and the road link costs $C=\{C_{ij}:L_{ij}\in L\}$ is a positive real-valued cost function $C:L\to R^{+}$. Let a vehicle trip from a source node s∈N to a destination node d∈N. Then, the user-equilibrium eco-routing computes the path *P*, which is a sequence of road links (i.e., P⊂L) for this individual vehicle, that minimizes the vehicle fuel consumption (i.e., $minimize\;\sum_{l_{ij}\in P}C_{ij}$). This shortest path problem can be easily solved using Dijkstra’s algorithm [16].

The eco-routing problem can be also solved using integer linear programming, as shown in Equation (Equation 1). The variable $x_{ij}$ is either 0 or 1, which identifies whether a link $l_{ij}$ will be included in the shortest path.
(1)$$\begin{array}{cc} \mathrm{minimize} & \sum_{ij\in N}x_{ij}C_{ij}\\ \mathrm{subject}\;\mathrm{to}\;:\; & \sum_{j}^{}x_{ij}-\sum_{j}^{}x_{ji}=\left\{\begin{array}{ccc} 1 & \mathrm{if}\;\;\;i=s\\ -1 & \;\;\;\mathrm{if}\;i=d & \forall \;i\in N;\;k\in F\\ 0 & otherwise. \end{array}\right. \end{array}$$

Both of these solutions are user-equilibrium models. A general disadvantage of user-equilibrium models for eco-routing is that it can result in overloading the best routes, which increases the system-wide fuel consumption. These user-equilibrium models try to overcome this problem by periodically updating the link costs and recomputing the best routes, consequently reducing the load on the previous best routes. But, on the other hand, these periodic updates and recomputation of routes produces route oscillation among the best routes.

A main question that arises here is how to compute the link costs $C_{ij}$. This is a major difference between different research efforts in this area. Some techniques use simplified mathematical macroscopic models to compute the link fuel cost, based on the average speed on the link (such as the model developed in [3]). Other techniques utilize data-driven macroscopic models, such as the model developed in [17]. Both of these techniques lack accuracy, because they can not capture the impact of the second-by-second speed and acceleration which are the dominant factors in computing fuel consumption. Consequently, in this paper, to achieve high fuel consumption computation accuracy we utilize a microscopic model, the VT-Micro [18], whose overview is presented in the next section.

## 3. FB-ECO Operation Assuming Ideal Communication Network

This section describes the eco-routing operation and implementation. Since we use the INTEGRATION software [15] as a simulation and modeling tool to conduct our study, this section starts by giving a brief overview of the INTEGRATION software and its eco-routing implementation as a user-equilibrium model using the shortest path technique.

### 3.1. INTEGRATION Software

The INTEGRATION software is an agent-based microscopic traffic assignment and simulation software [15]. It is capable of simulating large-scale networks, up to 10,000 road links and more than 500,000 vehicle departures at a time granularity of 0.1 s. This high time-resolution allows detailed analyses of many traffic theory phenomena, such as acceleration, deceleration, lane-changing, and car following behavior. It computes a number of measures of performance including delay, stops, fuel consumption, hydrocarbon, carbon monoxide, carbon dioxide, and nitrous oxide emissions, and the crash risk for 14 crash types. The details of the INTEGRATION software can be found in [15].

The INTEGRATION model has been developed over three decades, and has been extensively tested and validated against empirical data and traffic flow theory. Furthermore, the INTEGRATION software is the only software that models vehicle dynamics and estimates mobility, energy, environmental, and safety measures of effectiveness.

### 3.2. FB-ECO in INTEGRATION, Assuming Ideal Communication

Like most other traffic simulators, by default INTEGRATION assumes an ideal communication network (i.e., all packets are delivered correctly and with no delay). This subsection describes how eco-routng as a feedback system works assuming ideal communication, and the next section describes the communication model and how it was incorporated within the INTEGRATION eco-routing system.

In the INTEGRATION framework, eco-routing is developed as a feedback system that assumes the vehicles are connected and equipped with GPSs. Moreover, a vehicle is assumed to be capable of calculating the fuel consumption for each road link it traverses and communicating this information to the TMC. In INTEGRATION, the fuel consumption and emission rates of each vehicle are calculated every second, based on instantaneous speed and acceleration. As shown in Figure 1, every deci-second, the speed and the acceleration of each vehicle are calculated, as well as the vehicle’s fuel consumption rate.

Each vehicle accumulates this fuel consumption rate on each road link. Then, whenever the vehicle exits that link, it updates the link cost. Based on the ideal communication assumption, these updates are promptly added to the routing information in the TMC module.

### 3.3. Estimating Vehicle Fuel Consumption

The granularity of deci-second computations permits the steady-state fuel consumption rate for each vehicle to be computed each second, on the basis of its current instantaneous speed and acceleration level. INTEGRATION computes the fuel consumption and emission levels using the VT-Micro model, which is detailed in [18]. The VT-Micro model was developed as a statistical model from experimentation with numerous polynomial combinations of speed and acceleration levels to construct a dual-regime model, as demonstrated in Equation (Equation 2):(2)$$F\left(t\right)=\left\{\begin{array}{ccc} exp\left(\sum_{i=1}^{3}\;\sum_{j=1}^{3}L_{i,j}v^{i}a^{j}\right) & if & a\geq 0\\ exp\left(\sum_{i=1}^{3}\;\sum_{j=1}^{3}M_{i,j}v^{i}a^{j}\right) & if & a<0, \end{array}\right.$$
where $L_{i,j}$ are model regression coefficients at speed exponent *i* and acceleration exponent *j*, $M_{i,j}$ are model regression coefficients at speed exponent *i* and acceleration exponent *j*, *v* is the instantaneous vehicle speed in (km/h), and *a* is the instantaneous vehicle acceleration (km/h/s).

### 3.4. Updating the Cost Information

The vehicle route is a sequence of connected links. Thus, if a route $R_{n}$ consists of *k* links, the total route fuel consumption cost $F_{R_{n}}$ is the summation of the fuel consumption of the constituting links as expressed in Equation (Equation 3):(3)$$F_{R_{n}}\left(t\right)=\sum_{m=1}^{k}F_{l_{m}}\left(t\right),$$
where $F_{l_{m}}\left(t\right)$ is the fuel cost for the link $l_{m}$ at the current time.

Initially, $F_{l_{m}}$ is computed based on the free flow speed. Then, this value is updated based on the updates from probe vehicles. Based on the ideal communication assumption, whenever a vehicle exits a link $l_{m}$, INTEGRATION uses the reported vehicle’s fuel consumption $C_{m}$ on this link to update the link cost $F_{l_{m}}$. It uses a smoothing factor α as shown in Equation (Equation 4); a typical value of α in INTEGRATION is 0.2.
(4)$$F_{l_{m}}\left(t+1\right)=\left(1-\alpha \right)\;F_{l_{m}}\left(t\right)+\alpha \;C_{m}.$$

The link costs are updated upon the arrival of new updates. Then, the routing engine uses the latest link costs to periodically recompute the best routes.

## 4. FB-ECO with Realistic Communication Modeling

This section describes the operation of the FB-ECO with realistic communication model. If first gives a brief description of the communication model, then describes how this model is incorporated within the INTEGRATION software.

### 4.1. Modeling the VANET Communication

To study the impact of communication on FB-ECO, we utilize the communication model that we developed in [6]. The model has two main components: The Medium Access Control technique (MAC) component and the queuing component. A two-dimensional Markov chain is used to model the MAC, based on the IEEE 802.11p standard [1]. The M/M/1/K queuing model [19] is used to represent the queuing process in the MAC layer of each individual vehicle. The model uses the communication configuration inputs (such as average packet size, average background packet generation rate, communication speed, communication range, and the queue capacity) and the current network condition (such as vehicle density in the sender communication range and the vehicle’s connectivity to the RSUs) to compute a set of network performance parameters such as packet drop probability and packet delay.

One important advantage of this communication model over the previous ones is that it considers the MAC layer queue size and its impact on communication performance. For instance, the smaller the queue size, the lower the number of packets that can be queued, and consequently, in the case of high packet traffic rates, many of the packets will be rejected by the queue, which will increase the packet drop ratio. On the other hand, a larger queue size will result in increasing the queuing delay to very long delays. In contrast with the previous models, the model we developed assumes a finite queue size in the MAC layer, which enables the model to consider the queuing process. To build a realistic model, the M/M/1/K queuing model [19] is incorporated into the MAC protocol, so that the back-off technique and the queue interact with each other. Consequently, with the help of this queuing model, we were able to compute both the queuing and processing delay. In addition, the queuing model parameters were used to estimate the throughput and packet drop rate.

The model also supports both saturated and unsaturated data traffic conditions. So, it can be used at different packet generation rates.

The limitation of this model is that it assumes only one Access Category (AC) in the MAC layer, compared to four ACs in the IEEE 802.11p specifications. So, it does not support Quality of Services (QoS), which is supported in the IEEE 802.11p by enabling four ACs. The main purpose of this assumption is to simplify the model, in order to enable modeling of large-scale vehicular systems. This assumption is based on a comparative simulation study we made between a single AC and multiple ACs in VANET. This comparison showed that the performance of a single AC is similar to that of the Best Effort (BE) access category in the full fledged model.

The detailed description of the model, its assumptions, and derivation are described in details in [6]. However, for the sake of completeness, we will summarize it in the following two subsections.

### 4.2. MAC Representation and Solving the Model

In the IEEE 802.11p, when an AC has a packet to send, it initializes its back-off counter to a random value within a given range, called the contention window (CW). Then, it senses the medium until it becomes idle. If the medium continues to be idle for a specific time period called Arbitration Inter-Frame Space (AIFS), it counts down its back-off counter. When this counter becomes zero, the AC can send its frame. Within the same station, two ACs can start transmitting at the same time: This situation is known as internal collision. In this case, the higher priority AC will be granted the transmission while the lower priority AC will double its CW range, re-initialize the back-off counter, and back-off again.

If two stations start sending at the same time, the collision of the two signals will destroy both of the frames. So, when a station sends a frame, it has to wait for an acknowledgment (ACK). If an ACK was not received within a specific time period, a collision is assumed, and the station must double its CW range and try to re-transmit the frame.

Figure 2 shows the model that represents this system. A queue of size *K* packets is used to en-queue packets arriving from the upper layer. A two-dimensional Markov chain is utilized to represent the MAC process described above. State 0 in the Markov chain represents the system-empty state (both the system and the queue are empty). Each of the other states is defined by (i,j), where *i* and *j* are the back-off stages and back-off counter value, respectively. Table 1 shows the symbols used for this model.

To solve this model, we start from the Markov chain and derive all the state probabilities as a function of P(0,0). The summation of all state probabilities should be equal to 1. We incorporate the queuing parameters into the model.

From the Markov chain in Figure 2, we can compute the probability that the system is in state P(0,j). P(0,j) can be expressed as:(5)$$\begin{array}{c} P\left(0,j\right)=\frac{w_{0}-j}{w_{0}}\;\frac{1-q_{0}}{p_{idle}}\left(P\left(0\right)+P\left(M+f-1,0\right)+\left(1-p_{col}\right)\sum_{i=0}^{M+f-2}P(i,0)\right)\;j=1,2,\ldots ,w_{0}-1. \end{array}$$

And P(0,0) can be expressed as:(6)$$\begin{array}{c} P\left(0,0\right)=\left(1-q_{0}\right)\left(P\left(0\right)+P\left(M+f-1,0\right)+\left(1-p_{col}\right)\sum_{i=0}^{M+f-2}P(i,0)\right). \end{array}$$

From Equations (Equation 5) and (Equation 6), we can derive P(0,j) as:(7)$$P\left(0,j\right)=\frac{w_{0}-j}{w_{0}}\frac{1}{p_{idle}}P\left(0,0\right)\;\;\;j=1,2,\ldots ,w_{0}-1.$$

P(i,0) and P(i,j) can be calculated, as in Equations (Equation 8) and (Equation 9)
(8)$$P\left(i,0\right)=p_{col}^{i}\;P\left(0,0\right)\;i=0,1,\ldots .,M+f-1,$$
(9)$$\begin{array}{c} P\left(i,j\right)=\frac{w_{i}-j}{w_{i}}\frac{p_{col}^{i}}{p_{idle}}P\left(0,0\right)\\ i=1,2,\ldots ,M+f-1\;and\;j=1,2,\ldots ,w_{i}-1. \end{array}$$

Finally, we can compute P(0) as:(10)$$P\left(0\right)=\frac{q_{0}}{1-q_{0}}P\left(0,0\right).$$

Since the summation of all the probabilities equals 1, we have:(11)$$P\left(0\right)+\sum_{i=0}^{M+f-1}P\left(i,0\right)+\sum_{k=1}^{w_{0}-1}P\left(0,k\right)+\sum_{i=1}^{M+f-1}\sum_{k=1}^{w_{i-1}}P\left(i,k\right)=1.$$

Notice that the window exponential factor is α for i≤M; that is, $w_{i}=w_{0}\;\alpha^{i}\;\forall \;i\leq M$, and $w_{i}=w_{0}\;\alpha^{M}\;\forall \;i>M$. By plugging $w_{i}$ into Equation (Equation 11) and using some math, we can calculate P(0,0), as shown in Equation (Equation 12).
(12)$$\begin{array}{c} P\left(0,0\right)=(\frac{q_{0}}{1-q_{0}}+\frac{1-{p_{col}}^{M+f}}{1-p_{col}}+\frac{w_{0}-1}{2\;p_{idle}}+\frac{1}{2\;p_{idle}}[\left(\alpha^{M-1}\;w_{0}-1\right)\frac{{p_{col}}^{M-1}-{p_{col}}^{M+f}}{1-p_{col}}\\ +w_{0}\frac{\alpha \;p_{col}-{\left(\alpha \;p_{col}\right)}^{M-1}}{1-\alpha \;p_{col}}+\frac{p_{col}-{\left(p_{col}\right)}^{M-1}}{1-p_{col}}]{)}^{-1} \end{array}$$

To solve the model, we need to calculate the values of $p_{col},p_{idle}$ and $q_{0}$. To do that, we derive the relationship between these three parameters and the state probabilities.

A collision will happen when two or more stations start transmission in the same time slot. Let the probability that a station starts transmitting at a time slot be $p_{trans}$, then $p_{trans}=\sum_{i=0}^{M+f-1}P\left(i,0\right)$. When a station sends a packet, the probability that this packet collides is $p_{col}=1-{\left(1-p_{tran}\right)}^{N-1}$. Therefore, for the entire system, the medium will be idle at any time slot only if no station is sending: $p_{idle_{slot}}={\left(1-p_{tran}\right)}^{N}$. The station decides that the medium is idle ($p_{idle}$) after AIFS idle time slots in a row: $p_{idle}={p_{idle_{slot}}}^{AIFS}$. For the entire system, the probability that a packet is successfully transmitted without collision is $p_{suc}=N\;p_{tran}\;{\left(1-p_{tran}\right)}^{N-1}$.

### 4.3. Using the M/M/1/K Model

The only missing part to completely solve the model is finding a relation between $q_{0}$ and the state probabilities. To solve for $q_{0}$, we use the M/M/1/K model, where we assume the packet interarrival time, between packets, is exponentially distributed with an average rate λ. Assuming that the service rate is $\mu =\frac{1}{T_{serv}}$, where $T_{serv}$ is the average packet processing time. $T_{serv}$ is the summation of the average time the packet stays in each stage. The packet can stay a time $T_{w}$ in every state, plus the average frame transmission time $T_{tr_{av}}$, which can be calculated as:(13)$$T_{w}=p_{fail}\;T_{f}+p_{suc}\;T_{s}+\frac{1}{p_{idle}}T_{slot},$$
(14)$$T_{tr_{av}}=p_{col}\;T_{f}+\left(1-p_{col}\right)\;T_{s},$$
where $p_{fail}=1-p_{suc}-p_{idle_{slot}}$; $T_{s}$ and $T_{f}$ are the successful transmission time and the failed transmission time, respectively; and $T_{s}$ and $T_{f}$ depend on whether MAC uses the basic or advanced (RTS/CTS) access modes.

The term in $T_{w}$ is the time required by the station while sensing the medium to ensure it is idle. Consequently, the service time $T_{serv}$ can be calculated as:(15)$$T_{serv}=T_{slot}\sum_{i=0}^{M+f-1}\left(\frac{T_{w}\left(w_{i}-1\right)}{2}+T_{tr_{av}}\right){p_{col}}^{i}.$$

Now, we can calculate the traffic intensity $\rho =\frac{\lambda }{\mu }$, and subsequently $q_{0}$, as:(16)$$q_{0}=\left\{\begin{array}{cc} \frac{1-\rho }{1-\rho^{K+1}} & \lambda \neq \mu ;\\ \frac{1}{K+1} & \lambda =\mu . \end{array}\right.$$

Using these equations, we can solve this model and estimate the total communication drop profitability and delay.

### 4.4. The Communication Model Validation

The model shown an accurate estimation of both packet drop probability and packet delay at different communication settings, including different packet sizes and different packet generation rates. An example of the model validation results is shown in Figure 3 and Figure 4. These two figures compare the model output to the simulation output, resulting from the OPNET software [20] (which is a powerful communication networks simulation tool).

### 4.5. FB-ECO Operation with V2I Communication

In this paper, to model the communication we assume V2I communication setup. The communication model, described in the previous subsection, is implemented within the INTEGRATION software, and the behavior of the eco-routing is modified to adapt the communication parameters (packet drop probability and packet delay), as illustrated in Figure 5, which shows three modules working in parallel; mobility module that moves the vehicles, a communication module that computes the packet drop probability and delay, and an updating module which updates the link cost information. The two later modules (communication and updating modules) are two new modules we added, to the capture the impact of communication on eco-routing. The three modules work together as follows:

When a vehicle finishes a road link, instead of directly updating the link cost to the TMC (as shown in Figure 1), the FB-ECO module in this vehicle sends this information to the communication module by adding a new packet to the transmission queue. Then, while the vehicle moves, the communication module checks for the connectivity. If the vehicle is not connected to an RSU (i.e., there is no RSU in its communication range), the queue will not be processed and the packets in the queue will be held. Whenever the vehicle is connected to an RSU, the communication module will process the packets in the queue.

For each packet in the transmission queue, the communication module first calculates its drop probability $P_{drop}$ as described earlier. Then, it generates a uniformly distributed random number to find whether this packet should be correctly delivered. If the packet should be delivered, the communication module calculates its average total delay, and inserts it into a time-based ordered queue. Consequently, it will be processed by the updating module in its time of arrival, where the updating module uses Equation (Equation 4) to update the link cost.

In this way, the modified behaviour of the FB-ECO accounts for both the packet drop probability and packet end-to-end delay.

## 5. Simulation and Results

To achieve realistic results in our study, it is important to use a real network with real calibrated traffic. This section starts by describing these points. Then, the results, in the case of ideal communication, are presented in Section 5.2. In Section 5.3, the results for the realistic communication case are presented and compared to the ideal communication case.

### 5.1. Simulation Network and Traffic Calibration

The downtown area in the city of Los Angeles (LA), shown in Figure 6, is used for the simulation and analysis. The red points and surrounding circles are the RSU locations and their communication ranges, which are used in communication modeling (the allocation for the RUSs will be described later). This road network is about 133 Km${}^{2}$. It has 1625 road network nodes, 3561 road links, and 459 traffic signals. With regards to the vehicle traffic demand, we use a calibrated traffic demand, based on the vehicle count data from loop detectors in the same area. This data is collected from multiple sources, as described in detail in [21]. This traffic demand represents the morning peak hours in the downtown area of the city of LA, which continues for 3 h from 7:00 a.m. to 10:00 a.m. We added one hour for traffic pre-loading. So, the demand runs for four hours. However, we run the simulation for eight hours to give the vehicles enough time to finish their trips. To study the impact of different traffic origin-destination demand (OD) levels, the calibrated traffic rates are multiplied by OD Scaling Factors (ODSFs) ranging from 0.2 through 1.0 at a 0.2 increment, which produces 5 traffic demand levels. The total number of vehicles that are simulated in the full traffic scenario (ODSFs = 1.0) is more than 530,000 vehicles.

For each of these 5 traffic levels, we use 10 different penetration ratios of the probe connected vehicles, ranging from 0.1 through 1.0, at a 0.1 increment. Thus, we have to run 50 scenarios using the ideal communication configuration, and then rerun the same 50 scenarios with communication modeling enabled. The next two subsections present and analyze the results, in these two cases.

### 5.2. Ideal Communication Case

We start our analysis by running ideal communication scenarios, and then compute the average vehicle fuel consumption in each.

#### 5.2.1. The Importance of Feedback

Figure 7 shows the average vehicle fuel consumption for the 5 traffic demand levels at penetration ratios 0 through 1, at increments of 0.1. The figure shows the importance of the fuel consumption feedback for the system-wide performance, even at low traffic demand levels, where it shows that at ODFS = 0.4, increasing the penetration rate from 0.0 (no feedback) to 0.1 results in decreasing the average vehicle fuel consumption by about 12.4%. The results shows that at 0.0% penetration rate, the network gridlock happened at an ODSF = 0.6 and higher demand level, consequently some vehicles became stuck in the network. So, the fuel consumption shown in this figure for ODSF = 0.6 and higher are estimated, based on the vehicles that finished their trips.

#### 5.2.2. Impact of Penetration Rate on Fuel Consumption

Figure 7 also demonstrates that, with feedback enabled (penetration rate > 0) and at low traffic demand levels (ODSF = 0.2 and ODSF = 0.4), the penetration rate does not have a significant impact on fuel consumption. The reason for this is that, at low traffic demand levels, vehicles run almost at the free flow speed, and there are no significant changes in the network status (such as network congestion and associated increase in the link costs) that need to be updated at the TMC. In these two demand levels, the network is not congested where the average vehicle density is about 5 veh/km/lane and 9 veh/km/lane (in the cases of ODSF = 0.2 and ODSF = 0.4, respectively), as shown in in the network fundamental diagram in Figure 8 and Figure 9.

It is also clear in Figure 7 that, as the demand level increases, the lack for updates at 0.1 penetration rate results in increasing the fuel consumption, compared to higher penetration rates. This means that, as the demand level increases, the importance of having enough updates becomes higher to reflect the changes in the network status.

We notice also, in Figure 7, that at high traffic levels (ODSF = 0.8 and ODSF = 1.0), the system performance sometimes becomes worse when increasing the penetration rate. This is reasoned by the route oscillation effect of the shortest path dynamic routing techniques (bang-bang effect). What happens is that, if we have full and exact information about the network, then all the vehicles will take the best routes which results in the overloading of these routes (especially at high traffic demand levels), resulting in higher congestion on these routes and higher fuel consumption until the re-routing takes place. To avoid temporal oscillations in route choices, a white noise error function can be introduced into the link cost function. This allows vehicles to select slightly sub-optimum routes, if the costs along alternative routes are very similar and thus distribute the traffic. In our simulation, we introduce random white noise with a coefficient of variation equal to 0.05. Additionally, the low penetration rates implicitly introduce some other white noise to the link cost. Consequently, higher penetration rates sometimes produce worse performance.

Figure 10 shows the average fuel saving by increasing the penetration rate just from 0.1 to 0.2. It demonstrates that this saving is exponentially increasing with demand level, as shown by the trend line.

From Figure 7 and Figure 10, we can conclude that a penetration rate of 0.2 results in sub-optimal fuel consumption, which is near to the optimal fuel consumption in all cases. So, we can conclude that a 0.2 penetration rate for the probe vehicles is sufficient.

#### 5.2.3. Penetration Rate and Congestion Levels

To understand the results in more depth, we computed the network fundamental diagrams [22] for some of these scenarios. Figure 8, Figure 9, Figure 10, Figure 11 and Figure 12 compare these fundamental diagrams for different penetration rates, at different values of the ODFS.

Figure 8 and Figure 9 show that, at low traffic demand levels, the impact of the penetration rate is not significant. Meanwhile, Figure 11 shows the network fundamental diagram for ODSF = 0.8, where a penetration rate of 0.1 shows relatively high vehicles density combined with low traffic flow, which reflects temporary network gridlocks, that can be explained by the lack of routing information at a 0.1 penetration rate, which produces inaccurate routing decisions. The figure also shows that, for the higher penetration rates, starting at 0.2, the gridlocks do not exist. These gridlocks at the low penetration rate is reflected in the higher fuel consumption rates in Figure 7. This also supports our conclusion that 20% penetration rate is enough to deploy the FB-ECO systems and achieve acceptable performance.

Figure 12 shows that, at high traffic demand level (full demand rates), the impact of the penetration rate on network congestion becomes more significant. It shows that at a 0.1 penetration ratio, the network congestion becomes higher and the overall network enters the congested regime. It also shows that increasing the penetration ratio results in better system performance. It also shows that the overall network behaves the best at a penetration ratio of 0.6 or 0.8.

### 5.3. Impact of Realistic Communication on FB-ECO Performance

In this section, to study the impact on the communication performance on the FB-ECO, we use V2I communication by assuming that a set of RSUs are deployed in the network, and that vehicles communicate to the TMC through these RSUs. In V2I communication systems, it is necessary to allocate the RSUs in the network. The most economical method is to install the RSUs at traffic signals, to take the advantage of the existing connections and power sources. Figure 6 shows the allocation of the RSUs in the area of study. We selected 42 traffic signals from the network of 459 signals, using the greedy algorithm shown in Algorithm 1, that maximizes the network coverage with the minimum number of RSUs. The algorithm works as follows:
**Algorithm 1** Select traffic signals to install RSUs.1: **procedure**
Select traffic signals  ▹ Select the minimum number of traffic signals to install RSUs in such a way that maximizes the coverage2:    $S\leftarrow \{S_{i}:i=1,2\ldots .\}$3:    G←ϕ▹ The initial solution4:    **while**
S≠ϕ
**do**▹ There are uncovered signals5:        $C_{i}=\left\{S_{j}\in S:D_{i,j}<R_{Com}\right\}$▹ Recalculate the coverage6:        $Select\;S_{i}\in S\;that\;maximizes∥C_{i}∥$7:        $G\leftarrow G\cup S_{i}$8:        $S\leftarrow S\setminus C_{i}$9:    **return**
*G*▹ The selected signals

Here, it is assumed that the distance between the traffic signals $S_{i}$ and $S_{j}$ is $D_{i,j}$, and $C_{i}$ is the set of traffic signals covered by $S_{i}$. In other words, $C_{i}=\left\{S_{j}:D_{i,j}<R_{Com}\right\}$, where $R_{Com}$ is the communication range. The algorithm starts with *S* including all the traffic signals and an empty set *G* of selected signals. It calculates the coverage for each signal in *S*. Then, it selects the traffic signal that covers the maximum number of uncovered signals, adds it to the selected signals *G*, and removes it along with all the signals it covers from *S*. Steps 5 to 8 are repeated until *S* becomes empty. This algorithm does not guarantee coverage of the whole network. However, it covers the maximum number of signalized intersections with minimum cost (minimum number of RSUs).

After selecting these RSUs, we run the network using different traffic demands levels by using V2I communication modeling, in which the packets can be dropped and/or delayed. For each of the 5 ODSF scenarios, we run the 10 penetration ratios. In each case, the average fuel consumption per vehicle is calculated and compared to that in the ideal communication case. Figure 13, Figure 14, Figure 15, Figure 16 and Figure 17 show these results for ODSF values of 0.2, 0.4, 0.6, 0.8, and 1.0, respectively.

With regards to the VANET communication configuration, the V2I communication paradigm is used with a 1000 *m* communication range and a 50Packets/second background packet generation rate. The average packet size is set to 1000Bytes and the queue size in the MAC layer is set to 64Packets.

The figures demonstrate that, at lower traffic demand rates (ODSF = 0.2 and 0.4), the average vehicle’s fuel consumption rates in the ideal communication and realistic communication are similar (i.e., the impact of communication during these low traffic demands is not significant). However, as the traffic demand level increases, the average vehicle’s fuel consumption in the realistic communication case becomes significantly higher than that of ideal communication. We also notice that, at moderate traffic levels (0.6 and 0.8), the impact of communication is very small at low penetration ratios (0.1 through 0.3), and increases with the penetration ratio.

To understand these behaviors, we have to notice that, in general, the number of connected vehicles increases with the penetration ratio. Consequently, the packet drop rate increases with the number of connected vehicles, because of the increasing contention over the wireless medium.

At low ODSF values, the number of connected vehicles is very small, so the contention over the wireless medium is very limited. Consequently, the packet drop rate is very low. Thus, most of the link cost information is correctly delivered to the TMC, and the optimal routes are correctly computed. At moderate demand levels and low penetration ratios, the packet drop rates and delays are still acceptable. However, at these moderate demand levels, the drop rate increases with the penetration ratio to an unacceptable value, resulting in dropping most of the link cost updating packets. Thus, the information in the routing database at the TMC is insufficient to correctly represent the network state. Consequently, sub-optimal routes are used, which produces higher fuel consumption.

The results also demonstrate that, at high traffic demand levels, the average vehicle’s fuel consumption is significantly higher in the realistic communication cases, compared to the ideal communication cases at all the penetration ratios, as shown in Figure 18.

Figure 19 shows the packet average drop rates for different ODSF at different penetration ratios. It shows that, in highly congested cases, even low penetration ratios lead to high packet drop rates. These drop rates are reflected as sub-optimal routing decisions and higher fuel consumption, compared to the ideal communication, shown by the large difference between the two cases at all penetration ratios shown in Figure 17.

We notice in Figure 16 and Figure 17 that the curve trends are not fully consistent with this analysis: In some cases, increasing the penetration ratio produces significantly less fuel consumption. The reason for this is that, at these two demand levels (0.8 and 1.0), the sub-optimal routes result in network grid-locks ( these grid-lock happen at ODSF = 1.0 for all penetration ratios, and at ODSF = 0.8 for penetration ratios 0.3 and above) that cause some vehicles to become stuck in the network, and so do not finish their trips. The average vehicle fuel consumption, in these cases, is estimated based on the vehicles that finished their trips. These estimates are not accurate, because most of the vehicles that finished their trip did not experience the congestion or the gridlocks.

Figure 18 shows the percentage of increase in average fuel consumption per vehicle due to realistic communication modeling in all of the 50 scenarios. This percentage is computed as $100*\frac{F_{Real\_com}-F_{Ideal\_com}}{F_{Real\_com}}$, where $F_{Real\_com}$ and $F_{Ideal\_com}$ are the average fuel consumption in the realistic communication case and the ideal communication case, respectively. It can be easily concluded that the impact of communication is exponentially increasing with the vehicle traffic demand level at all the penetration ratios. It also shows that the percentage of fuel increases at ODSF equal to 0.8 and 1.0, and decreases in some cases (i.e., for penetration ratios 0.5 and above) because of the vehicles that become stuck in the network due to the grid-lock problem, which makes the fuel computation not accurate in these cases, as above.

Figure 19 shows the average packet drop rate in different scenarios. It is clear that the packet drop rate increases with both the ODSF and the penetration ratio, which is intuitive.

## 6. Conclusions and Future Work

In this paper, we conducted a sensitivity analysis to study the impact of the VANET communication system on the performance of an eco-routing technology. To do that, we developed a communication model for VANETs and modified the eco-routing navigation technique to account for the communication network performance, and to capture the impact of packet drop and delay on the feedback based eco-routing system. We ran this system on a real large-scale network, namely the downtown area of the city of Los Angeles. We also used a realistic calibrated traffic demand, using loop detector vehicle counts. This traffic calibration generated more than 530,000 vehicle trips in the network, during the morning peak period.

The simulation results showed a set of interesting conclusions:Under the ideal communication assumption, increasing the market penetration level resulted in improvements in the network-wide fuel consumption levels. However, market penetration levels between 20% and 30% gave acceptable performance.Using realistic communication modeling showed a trade-off when increasing the market penetration level. It showed that, at low penetration rates, the performance is acceptable because of the low packet drop rates. However, increasing the market penetration level results in increasing the fuel consumption because of the routing errors that occur, due to increased packet drop rates.The results showed that in both the ideal and realistic communication cases, FB-ECO can work properly at technology market penetration rates between 20% and 30%.The VANET communication network performance (packet drop and delay) can have significant effects on the dynamic eco-routing system performance, especially in highly congested networks. In some cases, it resulted in network-gridlock. This means that it is imperative to consider these impacts when deploying dynamic routing systems.

In this paper, we studied the impact of communication and market penetration rates on the user-equilibrium eco-routing model, based on the shortest path algorithm. Considering system-optimum routing is an interesting future research effort. An extension for this paper is to study the impact of different communication settings (e.g., different communication ranges, different background packet generation rates, and different packet sizes) on the system performance. Finally, it is imperative to study the temporal and spatial relation between the vehicular congestion and packet drop and delay. Such a study will give us a more in-depth understanding of these interactions and enable a prediction of the impacts of these parameters on the performance of eco-routing systems.

## Figures and Tables

**Figure 1 sensors-19-00290-f001:**
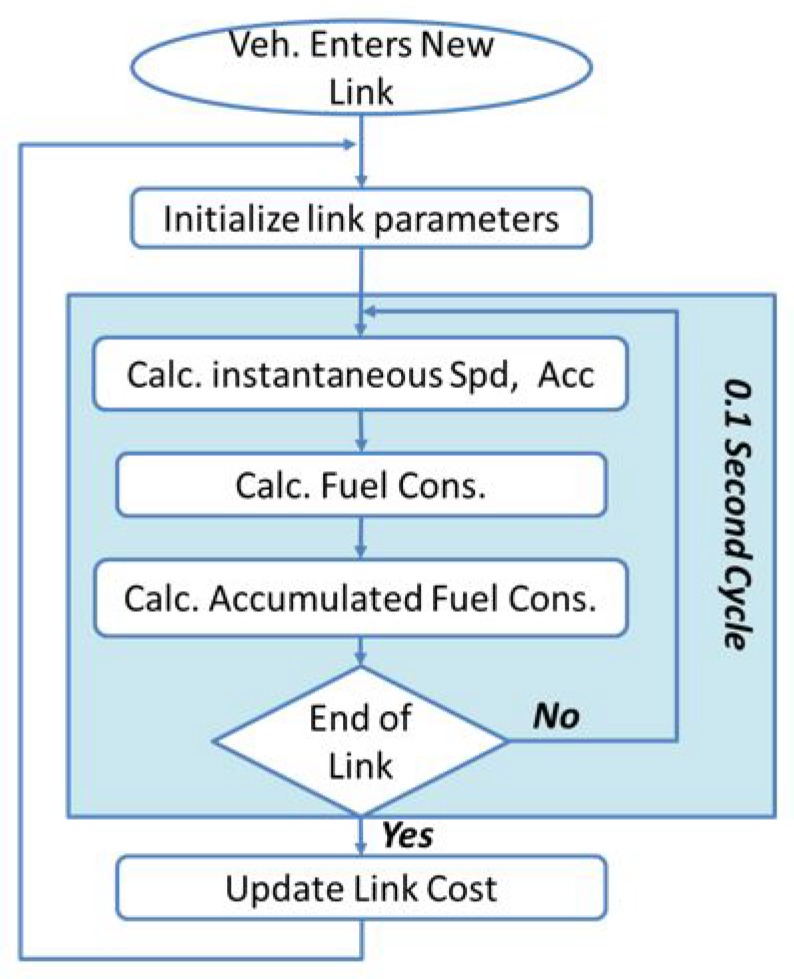
Eco-routing without communication modeling.

**Figure 2 sensors-19-00290-f002:**
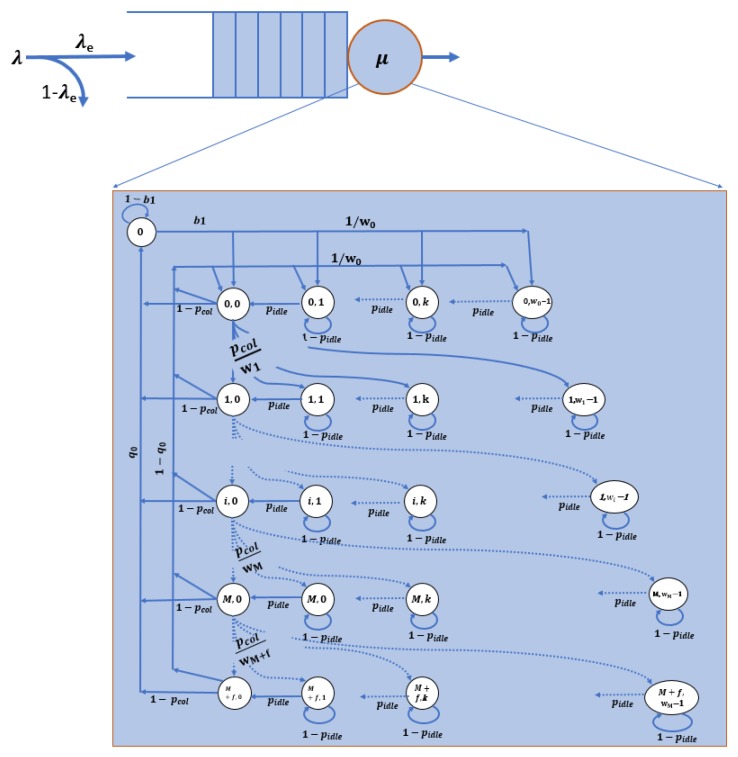
Markov chain model for the medium access.

**Figure 3 sensors-19-00290-f003:**
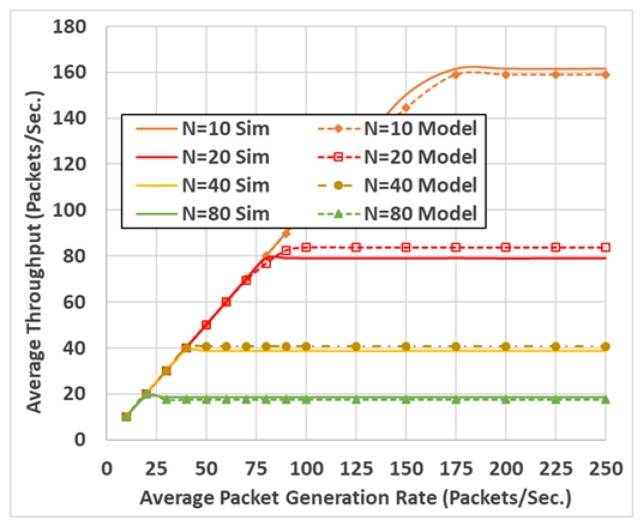
Average throughput per vehicle.

**Figure 4 sensors-19-00290-f004:**
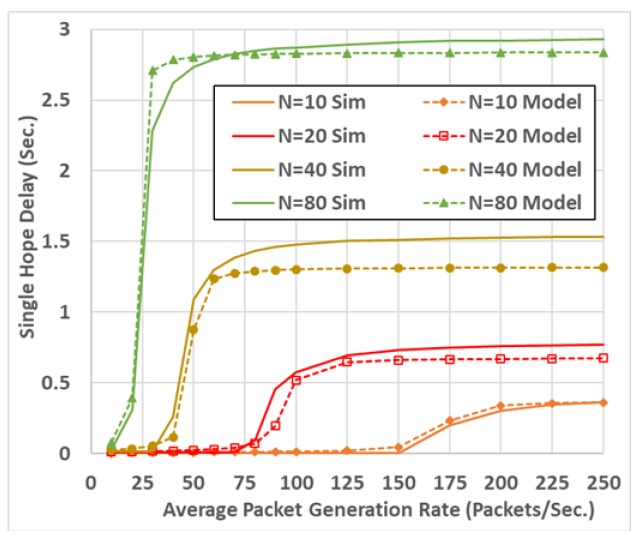
Average packet delay.

**Figure 5 sensors-19-00290-f005:**
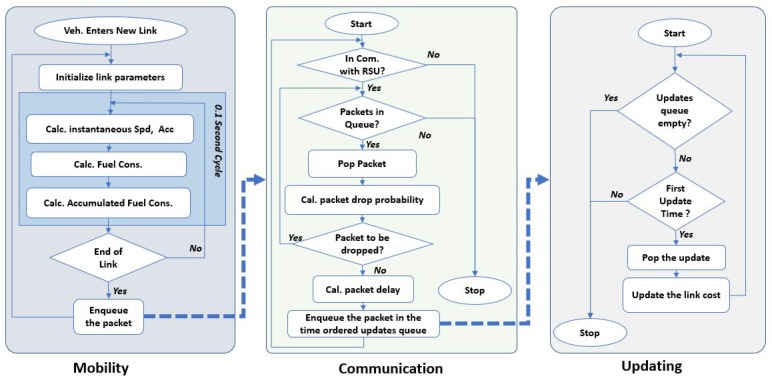
Eco-routing with communication modeling.

**Figure 6 sensors-19-00290-f006:**
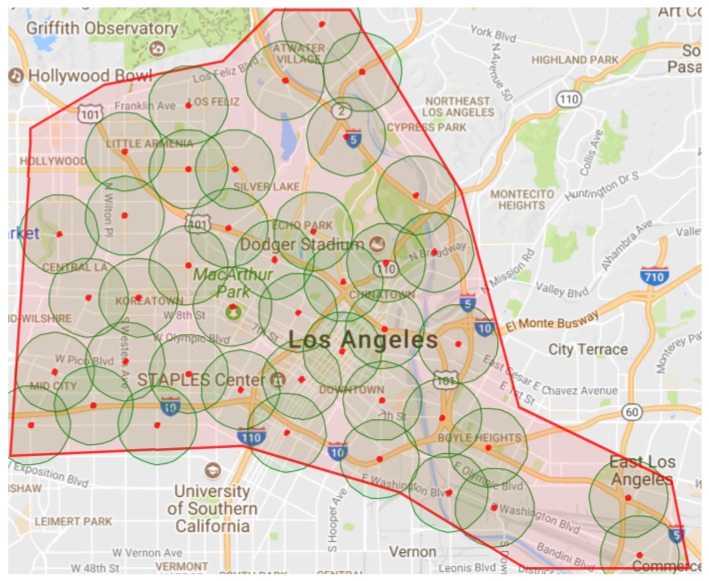
The Los Angeles (LA) downtown area and coverage map.

**Figure 7 sensors-19-00290-f007:**
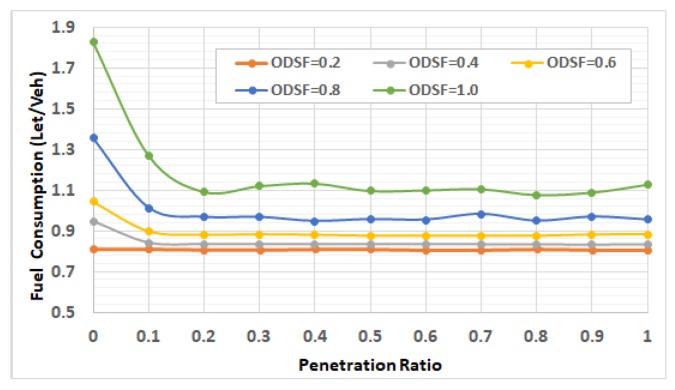
Impact of penetration rate on the average fuel consumption at different OD levels.

**Figure 8 sensors-19-00290-f008:**
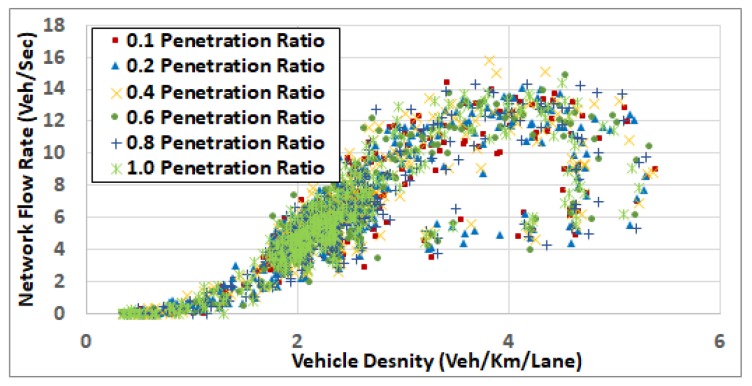
Network fundamental diagram at ODSF 0.2.

**Figure 9 sensors-19-00290-f009:**
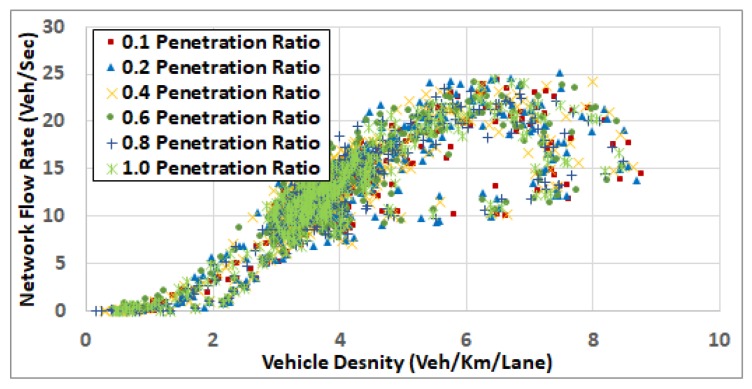
Network fundamental diagram at ODSF 0.4.

**Figure 10 sensors-19-00290-f010:**
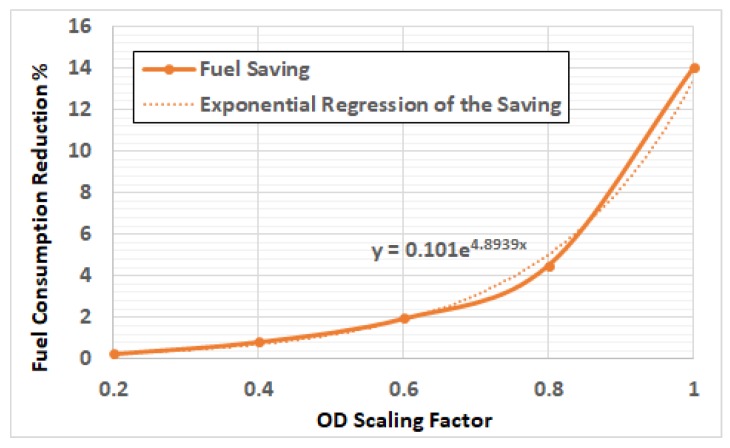
The average fuel saving by increasing the penetration rate from 0.1 to 0.2.

**Figure 11 sensors-19-00290-f011:**
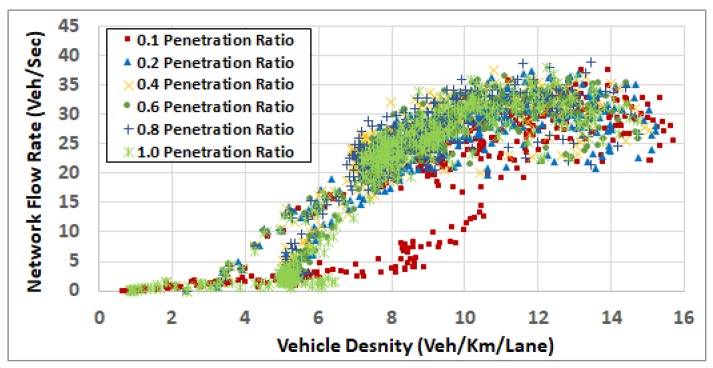
Network fundamental diagram at OD scaling factor 0.8.

**Figure 12 sensors-19-00290-f012:**
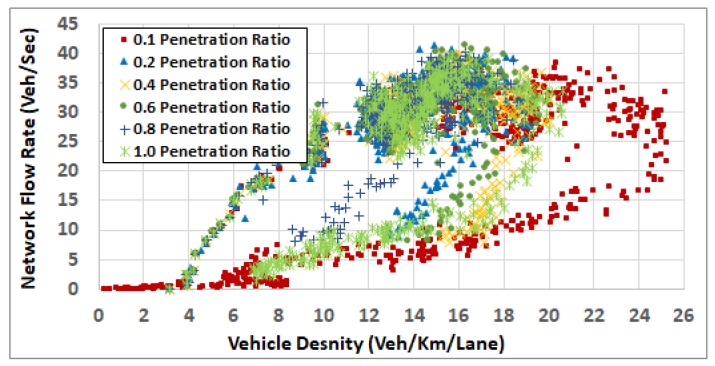
Network fundamental diagram at OD scaling factor 1.0.

**Figure 13 sensors-19-00290-f013:**
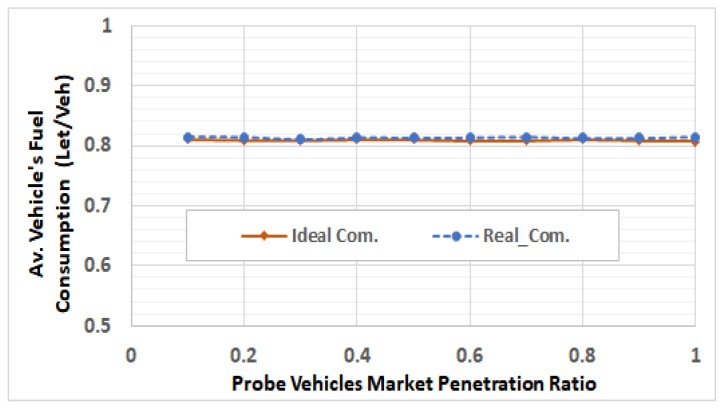
Average fuel consumption per vehicle at ODSF = 0.2.

**Figure 14 sensors-19-00290-f014:**
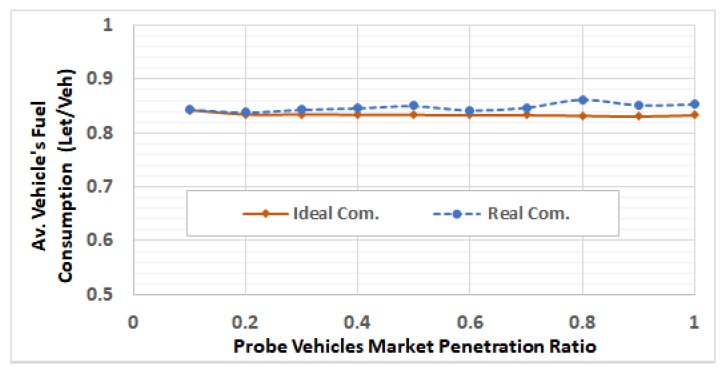
Average fuel consumption per vehicle at ODSF = 0.4.

**Figure 15 sensors-19-00290-f015:**
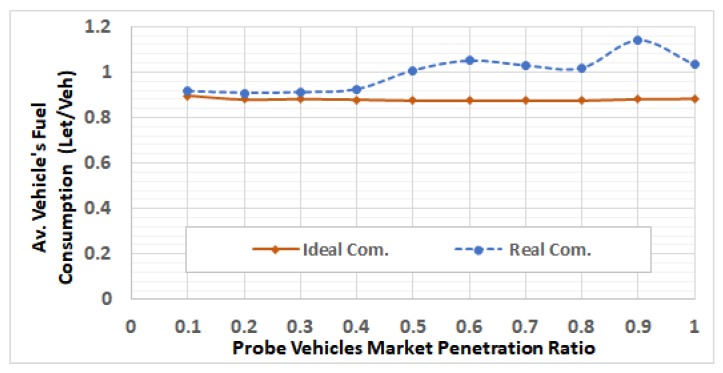
Average fuel consumption per vehicle at ODSF = 0.6.

**Figure 16 sensors-19-00290-f016:**
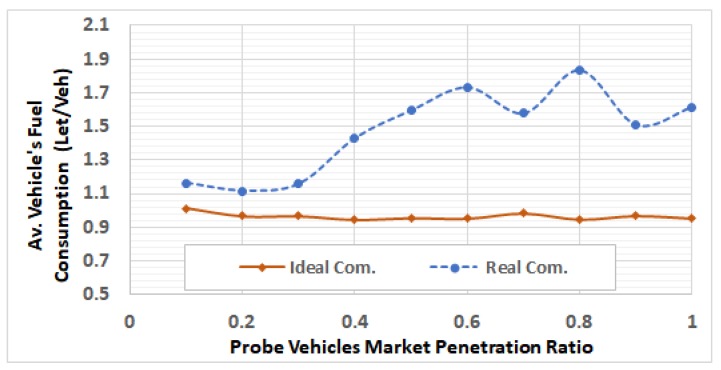
Average fuel consumption per vehicle at ODSF = 0.8.

**Figure 17 sensors-19-00290-f017:**
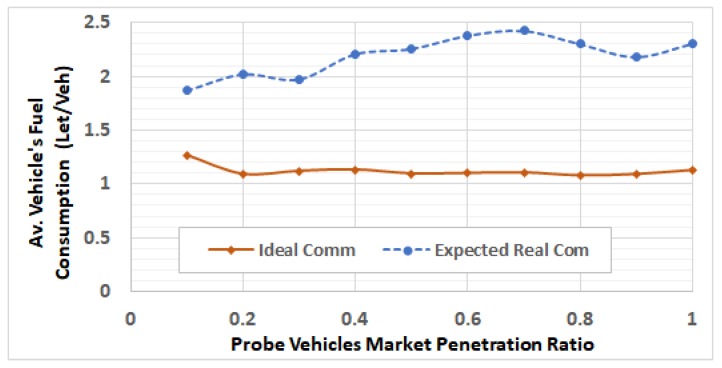
Average fuel consumption per vehicle at ODSF = 1.0.

**Figure 18 sensors-19-00290-f018:**
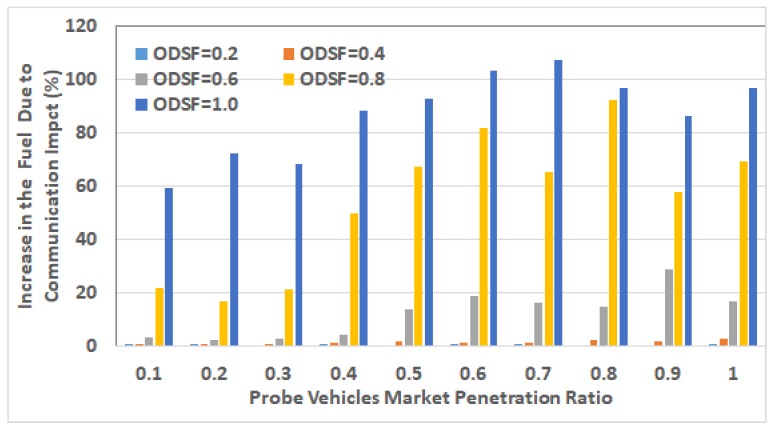
Impact of communication on the average vehicle’s fuel consumption.

**Figure 19 sensors-19-00290-f019:**
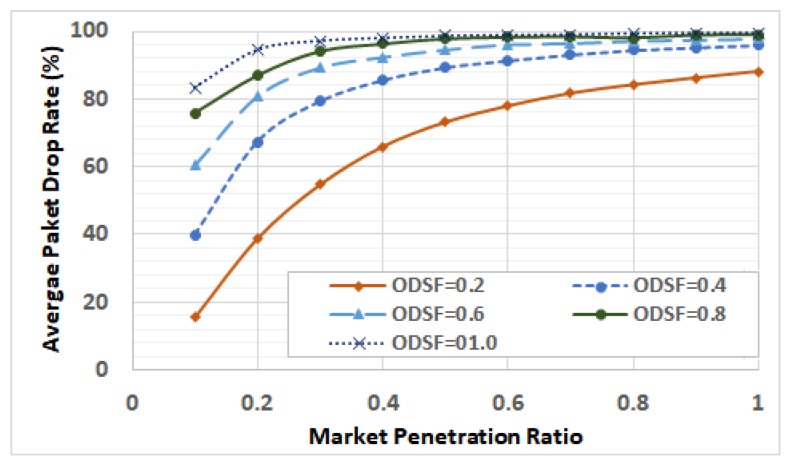
Average packet drop rates for different demand levels and penetration ratios.

**Table 1 sensors-19-00290-t001:** The model parameters.

Symbol	Description
*i*	The back-off stage number
*j*	The back-off counter
*M*	The maximum number of increases of the CW
*f*	The maximum number of retransmissions without increasing the CW
$w_{i}$	The CW range for stage *i*
$w_{0}$	The initial value for the maximum CW *i*
α	The CW increasing factor, where $w_{i}=w_{0}\alpha^{i}$. The typical value is 2.
$p_{idle_{slot}}$	The probability that a medium is idle in any time slot
$p_{idle}$	The probability that the medium is idle
$q_{0}$	The probability that the system is empty (no packet in the system)
$p_{suc}$	The probability that a medium is occupied with a successful transmission
$p_{fail}$	The probability that a medium is occupied with a failed transmission
$p_{tran}$	The probability that a station starts transmission in any time slot
$p_{col}$	The probability that the packet collides
P(i,j)	The probability that the system is in state (i,j)
λ	The packet arrival rate
μ	The packet service rate
$T_{serv}$	The packet service time
*N*	The number of vehicles in communication range
$T_{slot}$	The length of the time slot (sec)
$T_{s}$	The transmission time of a successful frame transmission
$T_{f}$	The transmission time of a failed frame transmission
ρ	The traffic intensity for the queuing model

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
