# Peer review of "City-Wide Eco-Routing Navigation Considering Vehicular Communication Impacts"

_sensors, 2019, doi:10.3390/s19020290_

Reviewer 1 Report

This paper analyzes the impact of VANET communication system on the FB-ECO navigation technique. Contributions of the paper are two-fold; the authors propose a communication modeling considering packet drop probability and delay for FB-ECO performance evaluation. Second, they include an updating phase that updates road link cost based on the updates from probe vehicles. Following on, different penetration ratios of probe vehicles are considered for the analysis. Also,  a proposal for RSU deployment is presented.  Comparisons have been made in results with the ideal communication and the proposed communication model. Work described in the paper is of good quality and addresses a major concern in the field of eco-routing technology. Nevertheless, I have the following concerns.

1)The VANET communication model is only mentioned and the reference (reference 6 ) is provided for readers interested in the detailed description of the model, its assumptions and its validation. I think that a brief summary of the model must be provided for the self-containing of the work. I suggest that the limitations of the model must be stated clearly for the best understanding of the paper. 

2)The same happens for the drop probability computation. A brief explanation must be included in the paper. Authors should take advantage that the journal does not have a limit of pages.

3) The influence of RSU deployment is not mentioned during the performance evaluation. It should be commented. I suppose that as the number of RSUs increases, the communication performance improves. 

4)In the results, an important penetration factor is not analyzed. I mean the 0% penetration factor (the current situation). Authors emphasize in the improvement for 0.1 to 0.2 penetration factor. However, it is important to know what happens when vehicles cannot compute updated shortest path because no one updates the road cost. This penetration factor must be included to highlight more the importance of penetration factor.

5)The Authors claims that a penetration factor of 0.2 is good enough. Nonetheless, their results show that under ideal communication, some higher penetration factors than 0.2 can improve the results (as can be expected )  but some others perform worse than 0.2 factor. An explanation must be provided. Why is there not a trend?. (See ODSF 0.8 and 1 for Fig 4) 

6) Results obtained for the communication model are very optimistic by considering a communication range of 1000 m. I recommend that authors change it to 500 m. They can also present both results. It would be even more interesting for readers.

Other minors comments

Line 257 - spelling mistake ‘network’

Line 261 - spelling mistake ‘one hour’

Lines 283,395,384 – Duplicated ‘the the’ is wrong at many places.

Lines 355, 360 – Figure 14 mentioned instead of 15

Line 386 – spelling mistake ‘Los Angeles’

Figures 6 to 9 -  x label needs revision “Vehicle Desnity ( Veh/km…./lane)”

Overall the paper is well-written and well-structured.

Author Response

Thank you for reviewing our paper, your comments are very valuable and important to improve the paper.

·        Comments 1 and 2:

We added the communication model description in addition to its advantages over the previous models and its limitations as well in subsections 4.1 through 4.4.

Reviewer 2 Report

The paper is well written and addresses an interesting topic. In my view there are no issues, and then it could be published in the present form

Just one minor comment: A communication range of 1000 meters is the maximum value that can be obtained with the technology used. It would be more realistic to use a smaller value (700-900 m), is your approach robust with respect to this issue?

Moreover I found some small errors in the text, reported hereafter:

pg2, lane 60: missing an "e" In addition to th impact

pg 9 lane 283 check the word “the” used two times

pg 10 lane 329:  check the word “the” used two times

pg 12 lane 384 check the word “the” used two times

pg 13 lane 393 check the word “the” used two times

Author Response

Thank you for reviewing our paper, your comments are very valuable and important to improve the paper.

·        The communication range

With regards to the communication range, we agree with your comment which is completely correct, 1000 m is the maximum distance specified in the IEEE802.11p standard.  In this paper we focus on the impact of the level of penetration rate on the system performance.

In our future plan, we will study in depth the impact of the different communication ranges (from 200 meters to 1000 meters) and different RSU locations, which is also related to the communication range. 

Additionally, in this paper, we assume an ideal communication channel model. While, realistically, the signal is subjected to attenuation, scattering and distortion. We also plan to study the probabilistic channel model on the performance of the FB-ECO.

·        The spelling errors

All the spelling errors have been corrected.

Round  2

Reviewer 1 Report

All my comments have been addressed. Thank you